# New Benzothiazole–Monoterpenoid Hybrids as Multifunctional Molecules with Potential Applications in Cosmetics

**DOI:** 10.3390/molecules30030636

**Published:** 2025-01-31

**Authors:** Desislava Kirkova, Yordan Stremski, Maria Bachvarova, Mina Todorova, Bogdan Goranov, Stela Statkova-Abeghe, Margarita Docheva

**Affiliations:** 1Agricultural Academy, Tobacco and Tobacco Products Institute, 4108 Markovo, Bulgaria; desislavaa894@gmail.com (D.K.); margarita_1980@abv.bg (M.D.); 2Department of Organic Chemistry, University of Plovdiv “Paisii Hilendarski”, 24 Tsar Asen Str., 4000 Plovdiv, Bulgaria; bychvarova@uni-plovdiv.bg (M.B.); minatodorova@uni-plovdiv.bg (M.T.); stab@uni-plovdiv.bg (S.S.-A.); 3Department of Microbiology and Biotechnology, University of Food Technologies, 26 Maritza Boulevard, 4002 Plovdiv, Bulgaria; b_goranov@uft-plovdiv.bg

**Keywords:** *α*-amidoalkylation, molecular hybridization, thymol, carvacrol, in silico predictions, sun-protection factor, PBSA, antimicrobial activity

## Abstract

The *Thymus vulgaris* and *Origanum vulgare* essential oils (contained thymol and carvacrol in a range of 35–80%) are used in various products in the fields of medicine, cosmetics, and foods. Molecular hybridization between benzothiazole (BT) and phenolic monoterpenoids is a promising method for the development of biologically active compounds. New benzothiazole–monoterpenoid hybrids were synthesized through a regioselective *α*-amidoalkylation reaction of thymol and carvacrol with high yields (70–96%). This approach is both simple and cost-effective, employing easily accessible and inexpensive reagents to produce target molecules. The structure of the synthesized compounds was characterized spectrally using ^1^H-, ^13^C-NMR, FT-IR, and HRMS data. The newly obtained compounds are structural analogues of the UVB filter PBSA, which is used in cosmetics. The spectral properties of the aromatic products thymol hybrid (2-(4-hydroxy-5-isopropyl-2-methylphenyl)benzo[*d*]thiazole) and carvacrol hybrid (2-(4-hydroxy-2-isopropyl-5-methylphenyl)benzo[*d*]thiazole) were successfully examined, using a validated spectrophotometric method. SPF values varied from 31 to 36, compared to the PBSA (30), and were observed at concentrations of 1–0.25 mM. 2-Hydroxyphenylbenzothiazoles are known antimicrobial and antioxidant agents that have potential applications in the food industry and cosmetics as preservatives and antioxidants. In this context, antimicrobial activity of the hybrid compounds was evaluated using the agar diffusion method against *E. coli*, *S. aureus*, *P. aeruginosa*, and *C. albicans*. Compounds of methyl-2-(4-hydroxy-2-isopropyl-5-methylphenyl)benzo[*d*]thiazole-3(2*H*)-carboxylate containing carvacrol fragments showed high activity against *Staphylococcus aureus* ATCC 25923 (with 0.044 μmol content). The radical scavenging activity was determined using ABTS and DPPH assays, the highest activity was exhibited by the thymol hybrids ethyl-2-(4-hydroxy-5-isopropyl-2-methylphenyl)benzo[*d*]thiazole-3(2*H*)-carboxylate (IC_50_—133.70 ± 10 µM) and methyl-2-(4-hydroxy-5-isopropyl-2-methylphenyl)benzo[*d*]thiazole-3(2*H*)-carboxylate (IC_50_—157.50 ± 10 µM), defined by ABTS. The aromatic benzothiazole–monoterpenoid hybrids are classified using in silico analyses as non-mutagenic, with low toxicity, and they are non-irritating to the skin. These compounds were identified as new hit scaffolds for multifunctional molecules in cosmetics.

## 1. Introduction

The synthesis of multifunctional hybrid molecules is a relatively new strategy, initially employed to overcome the increasing rate of drug resistance to anticancer or antitubercular agents [1,2]. This strategy involves the combination of various heterocycles, which could enhance the anticancer activity of the hybrid molecules compared to that of the starting compounds [3]. Additionally, molecular hybridization can be used to obtain new multifunctional molecules with potential applications in cosmetics. In recent years, various authors have reported new biologically active compounds, structural analogues of natural substances. For instance, Valverde Sancho et al. proposed a convenient method for obtaining hybrid molecules with proven high biological activity based on essential oils [4].

The interest in the application of essential oils (EOs) in organic synthesis is due to their perception as safe at certain concentrations, owing to their high volatility and biodegradability, as well as their incorporation into pharmaceuticals and agricultural products. Thyme EO is one of the widely used oils in various products in the fields of medicine. Additionally, it is used in the food industry and cosmetics as a preservative and antioxidant [5,6]. In traditional medicine, wild thyme (*Thymus serpyllum* L.) is included in a large number of herbal medicinal products, such as syrups, tinctures, teas, and decoctions [5]. Thyme EO is preferred for use in pharmaceuticals due to its high thymol content, which has strong antitussive, spasmolytic, and expectorant effects. Its antiseptic action is estimated to be 25 times more effective than phenol (PhOH), but with lower toxicity [7]. 

There are numerous types and varieties of thyme (*Thymus vulgaris*). The genus Thymus is one of the most important of the *Lamiaceae* family, represented by 214 species and 36 subspecies. In Europe, six genetically distinct chemotypes of thyme predominantly exist: two phenolic (thymol, carvacrol) and four non-phenolic (linalool, geraniol, *α*-terpineol, and trans-4-thujanol/4-terpinenol) [8]. The content of the monoterpenoids thymol and carvacrol in some essential oils reaches up to about 80% [9]. Thymol (2-isopropyl-5-methylphenol) and carvacrol (5-isopropyl-2-methylphenol) are the most important volatile organic compounds in thyme essential oil [7]. They are relatively unstable to oxidation, distillation, heating, or exposure to light [9,10]. Thymol and carvacrol are natural phenolic isomers with a wide range of pharmaceutical activities, including antibacterial, antimicrobial, antioxidant, cytotoxic, and insecticidal properties [10,11]. The literature represents plenty of data, including comparisons between the activities of thymol and carvacrol. Aazza and colleagues found that the antioxidant activity of the two isomers is approximately equal, while the anti-acetylcholinesterase activity depends on the position of the hydroxyl group relative to the two alkyl substituents [12]. Miladi et al. investigated the in vitro antibacterial activity of thymol and carvacrol against Gram-positive and Gram-negative bacteria. The results showed that thymol and carvacrol possess high activity against the tested strains [13]. 

The anticancer activity of carvacrol has been evaluated in vitro. It was found that carvacrol significantly reduces the viability of gastric adenocarcinoma (AGS) cells at concentrations ranging from 10 to 600 μmol/L [14]. In 2020, Mari and colleagues investigated cell cycle arrest and apoptosis in breast cancer cells using carvacrol [15]. A comparison of the anticancer effects of thymol and carvacrol was made by Elbe and colleagues. It was found that thymol is more effective than carvacrol in preventing the growth of ovarian cancer cells in a dose- and time-dependent manner, although the exact mechanism is still unclear [16].

The application of thymol and carvacrol in skincare products is due to its ability to protect cells from damage caused by oxidative stress, and it is also effective as an inhibitor of the enzymes elastase and tyrosinase [17,18,19]. Tyrosinase inhibitors are important in medicine and cosmetics due to their ability to suppress the overproduction of melanin and prevent unwanted effects on the skin [10]. Improper or excessive exposure to ultraviolet rays can lead to oxidative stress caused by excessive accumulation of reactive oxygen species (ROS) that are not balanced by effective antioxidant protection. Additionally, this is the main cause of both melanoma and non-melanoma skin cancer. Ultraviolet rays are classified as one of the main causes of skin cancer [20,21]. Many plant polyphenols, including flavonoids, are reported to have significant photoprotective effects on the skin (antioxidant, anti-inflammatory, and anticancer potential), and some also act as mild UV filters. The existence of hydroxyl groups and aromatic structures facilitates the absorption of UV radiation across a broad spectrum of wavelengths that underpins their characteristics [22]. Recent studies indicate that polyphenols can provide an effective source for protecting the skin from the harmful effects of UV rays (UVA and UVB) [23]. 2-Phenyl-1*H*-benzimidazole-5-sulfonic acid (PBSA) is widely used in cosmetic sunscreens due to its ability to absorb UVB rays and its high water solubility [24]. A recent work describes the application of PBSA in various cosmetic products and a series of its 2-heteroaryl-benzoazole analogues as privileged structures for new multifunctional molecules [25].

In 2020, our synthetic group successfully applied a multicomponent reaction for the amidoalkylation of hydroxyarenes toward the formation of 2-(hydroxyaryl)benzothiazolines and an aromatic product, 4-(benzo[*d*]thiazol-2-yl)phenol (BT-PhOH hybrid, Figure 1), with antibacterial properties [26]. The antibacterial activity of these molecules was investigated against two Gram-positive strains (*Staphylococcus aureus* ATCC 6538 P and *Bacillus licheniformis* ATCC 14580) and one Gram-negative strain (*Escherichia coli* ATCC 8739). The BT-PhOH hybrid was found to exhibit high antibacterial activity against *S. aureus* ATCC 6538P, with a MIC of 0.0063 mg/mL [26].

Djuidje and collaborators also introduced strategies for the synthesis of the aforementioned compound 4-(benzo[*d*]thiazol-2-yl)phenol and various other analogues (Figure 1) via condensation reaction between 2-aminothiophenols and different benzaldehydes. The synthesized compounds were evaluated for their multifunctional effectiveness as antioxidant, sunscreen filter, antifungal, and antiproliferative agents [27]. 2-Phenylbenzo[*d*]thiazole was chosen by Djuidje as a reference standard for appropriate discussing of structure–activity relationship, as the structures of 2-(benzo[*d*]thiazol-2-yl)benzene-1,4-diol and 5-(benzo[*d*]thiazol-2-yl)benzene-1,2,4-triol (Figure 1) were identified by the authors as molecules with high radical scavenging activity and potential for designing multifunctional drugs. C-2-substituted arylbenzoazoles can be highly effective as *β*-amyloid-targeting imaging agents; calcium channel antagonists; chemiluminescent, antitubercular, antiparasitic, and antitumor agents; and also photosensitizers [28,29,30,31]. The discovery of sunscreen products with combined additional antimicrobial and antioxidant properties is of great interest.

From some synthetic perspectives, the production of benzothiazole–monoterpenoid hybrid molecules with a carbon–carbon-linked benzoazole fragment, using readily available renewable sources, such as essential oils, is of great interest. The literature contains successful data on *O*-alkylation, *O*-acylation [31,32], and *C*-acylation [10] for synthesizing thymol and carvacrol derivatives. However, the data show that this is not easily applicable to *C*-alkylation processes. Here, we successfully applied a convenient two-step approach for *C*-alkylation of thymol and carvacrol for the obtaining of novel compounds with benzothiazole–monoterpenoid fragments. 

Reactions of α-amidoalkylation have played a crucial role in in our previous research for constructing carbon–carbon bonds [26,33,34]. Furthermore, the application of a multicomponent reaction (MCR) of amidoalkylation to modify natural compounds by introducing biologically active heterocyclic fragments to obtain new hybrid molecules is substantially observed [33]. Hybrid molecules containing a natural monoterpenoid in their structure are difficult to obtain through classical condensation reactions due to the limited availability of the starting aldehydes. Additionally, their sensitivity and tendency toward undesirable side reactions, such as oxidation, polymerization, or isomerization, represent a big challenge for organic synthesis. In this regard, combining of natural phenolic compounds with a BT fragment via multicomponent manner, and investigating the potential biological activity of the obtained hybrids would expand the scope of the intermolecular *α*-amidoalkylation reaction.

The focus of the present study is the application of this convenient method toward multifunctional molecules containing a natural fragment in their structure. Additionally, the reaction has the potential to expand chemical diversity in the design of new bioactive compounds. This study may open new opportunities for the synthesis of important and effective bioactive molecules by applying multicomponent reactions for the amidoalkylation of natural phenolic monoterpenoids.

## 2. Results and Discussion

### 2.1. Synthesis of Benzothiazole–Monoterpenoid Hybrids

#### 2.1.1. Amidoalkylation of Natural Monoterpenoids—Thymol and Carvacrol

Our synthetic methodology is based on our previously described application of multicomponent reaction for the amidoalkylation of quercetin [33], resorcinol [34], and other phenols [26] using *N*-acyliminium reagents derived from benzoazoles with alkyl chloroformates. As a suitable continuation, here we successfully amidoalkylated the natural monoterpenoids, thymol and carvacrol, according to Figure 1.

In this study, it was found that the selected monoterpenoids are more reactive nucleophiles compared to phenol (PhOH). The differences in the conditions for the amidoalkylation of the natural monoterpenoids thymol and carvacrol compared to the phenols and naphthols [26] are due to the presence of two alkyl groups (isopropyl and methyl), which further assist in the orientation during the amidoalkylation process. Additionally, both isomers are significantly more lipophilic than simple phenols. It was found that in all cases, the reaction proceeded regioselectively, with high yields ranging from 82% **4c** to 96% **4a** (Table 1). The substitution in the para-position relative to the hydroxyl group in the thymol and carvacrol fragment was confirmed by proton NMR spectra recorded at 80 °C in DMSO-d_6_. Analytically pure samples of only para-substituted products were isolated by column chromatographic separation on silica using a mixture of petroleum/diethyl ether as eluents. The purity was monitored by thin-layer chromatography (TLC) and confirmed by ^1^H-, ^13^C-NMR experiments (see Appendix A). For example, in the spectrum of compound **5a**, the singlet for the proton of the hydroxyl group is observed at 8.93 ppm, and the singlets for the two protons of the benzene ring of the carvacrol are assigned at 6.97 ppm and 6.74 ppm. Due to the observed rotameric effects at room temperature, it was necessary to record the NMR spectra at a higher temperature (80 °C) in DMSO-d_6_. Thus, averaged signals were observed, successfully confirming those which are expected for the newly obtained products. The structure of the newly obtained products 4, 5, 6, and 7 was identified by spectral data (see Appendix A). The biological activity of the tested compounds was examined without the need for their further purification.

(a)Effect of Solvent Polarity

The use of acetonitrile as a solvent for the synthesis of compound **4a** resulted in the reaction occurring over 24 h at room temperature, whereas the reaction with the same acyliminium reagent and phenol with the same reactant ratio (2:1) required 80 h [26]. We found that dry dichloromethane as a solvent reduced the reaction time to 6 h (Table 2). Reducing the excess of benzothiazole from 100% to 30% extended the reaction time to 24 h. In all cases, the multicomponent reactions of amidoalkylation proceeded regioselectively, yielding a single monosubstituted product with para-substitution relative to the phenolic group. The yields and reaction conditions for the amidoalkylation reactions of the natural terpenoids thymol and carvacrol are presented in Table 1.

(b)Effect of Alkyl Chloroformates

The dependence of the rate of the amidoalkylation reaction on the structure of alkyl chloroformates, observed in earlier studies [26], was confirmed. The N-acyliminium adduct derived from benzothiazole and 2,2,2-trichloroethyl chloroformate (Troc-Cl) was established to be the most reactive, and the reaction with thymol was carried out for 1 h with 30% excess of the heterocycle (BT) for the synthesis of **4c**. Additionally, the acyliminium reagent with methyl chloroformate was more reactive than that obtained with ethyl chloroformate according to the shorter reaction time period toward **4b** (Table 1). At the same time, products **4a** and **5a** were obtained with higher yields than **4b** and **5b** (Table 1), an outcome which is likely due to the higher stability of the acyliminium reagent with ethyl chloroformate. The amidoalkylation reaction using Troc-Cl proceeded faster, 1.5 h for carvacrol, with reactant ratio N-acyliminium reagent to natural terpenoid of 1.3:1. The yields of the newly synthesized products are 82% for **4c** and 86% for **5c**, as presented in Table 1.

(c)Effect of the Alkyl Substituent in Phenolics to Formation of C-C Bond in BT Ring

Since thymol and carvacrol are positional isomers with respect to the phenolic group, their nucleophilicity and solubility are similar, and the amidoalkylation reactions are expected to proceed analogously (in terms of the reaction yields), with para-substitution relative to the phenolic group. However, the results showed that reactions with carvacrol required significantly longer time periods: product **4a** was obtained in 6 h (yield 96%), while **5a** was obtain in 24 h, with a yield of 93%. This tendency was also observed in all examples (Table 1). The differences in reaction times, under the same conditions, are likely due to steric hindrance from the bulkier isopropyl group in the ortho-position of the electrophilic attack of the acyliminium reagent to carvacrol.

#### 2.1.2. Experimental Results of Oxidative Rearomatization

Two new aromatic hybrids based on benzothiazole—thymol and carvacrol compounds **6** and **7**—were obtained through an oxidative rearomatization reaction according to Figure 1. The oxidation reaction of thymol hybrids **4a**–**c** with two equivalents of the oxidizing agent 2,3-dichloro-5,6-dicyano-1,4-benzoquinone (DDQ) was investigated. The reaction time depends on the type of solvent. In dichloromethane, the reaction proceeded at room temperature for 30 min, while in acetonitrile, a longer reaction time of 2–3 h is required, leading to higher yields of product 6–70% (from **4a**), 71% (from **4b**), and 56% (from **4c**). It was found that during the reaction in acetonitrile, the product precipitates, allowing its isolation by filtration. The oxidation reaction of carvacrol hybrids **5a**–**c** was successful under the same conditions. Here, the reaction again proceeded at room temperature for 2–3 h, yielding product **7** with good yields—92% (from **5a**), 82% (from **5b**), and 41% (from **5c**), as described in Table 3.

#### 2.1.3. Synthesis of Compounds **4a**,**b** and **5a**,**b** from Essential Oils of Thyme (*Thymus vulgaris*) and Oregano (*Origanum vulgare*)

The widespread use of thyme and oregano essential oils in medicine, cosmetics, and the food sector facilitates the extraction of its primary constituents as isolated pure compounds. Carvacrol, which we use in synthetic reactions, is such a natural isolate, unlike synthetic thymol. We are interested in the possibility of obtaining the studied benzothiazole–monoterpenoid hybrids using thyme or oregano essential oil. In this regard, several EOs were purchased from the commercial suppliers—thyme with chemotype ‘Standard Winter’ and oregano with carvacrol chemotype. The tested chemotypes have an ISO standard, according to which the thymol content should be a minimum of 35% and a maximum of 55% [35], while for carvacrol, it varies in the range from 60% to 80% [36]. To confirm the content of the commercial product, a GC-MS/MS analysis was applied using a known methodology [37]. The determined thymol content in the thyme essential oil is 53.54%, with a retention time (RT) of 40.71 (Appendix A). The carvacrol content was established as 77.78%, with a retention time (RT) of 41.63 (Appendix A). The obtained results confirm that the component composition of tested EOs meets the requirements of the ISO standard.

In the search for optimal conditions for the synthesis of compounds **4a** and **4b**, it was found that N-acyliminium reagents were stable in the presence of thyme essential oil in dry dichloromethane, and the amidoalkylation reactions proceeded successfully over 24 h. In contrast to reactions involving pure thymol, diverse ratios of BT to the essential oil were employed, with thymol consistently present in excess. When using 0.600 g of the oil and 2 mmol of BT, the yields ranged from 28% to 35%. By increasing the amount of oil to 1.2 g (with thymol content of 4.3 mmol), the highest yields were obtained: 55% for **4a** and 62% for **4b**. For the comprehensive evaluation of this approach, compounds **5a** and **5b** were successfully synthesized in 55% for **5a** and 60% for **5b** yield, respectively, with 0.488 g (with carvacrol content of 2.5 mmol) of oregano oil for 24 h at room temperature.

These studies demonstrated the potential application of essential oils for the synthesis of new biologically active molecular hybrids.

### 2.2. Biological Activity

#### 2.2.1. In Silico Predictions of Benzothiazole-Based Hybrids Toxicity

In silico toxicology is a rapidly developing field concerning the safety of chemical substances and the prediction of their toxicity [38]. Quantitative structure–activity relationship (QSAR) uses statistical methods to analyze the mathematical relationships between the chemical structure of a substance and its biological or toxicological properties. Many studies focus on developing QSAR models that can predict the mutagenicity of chemical compounds through Ames Mutagenicity, as well as other toxicological endpoints such as carcinogenicity and general toxicity [39,40,41,42]. The application of various computer models aims to reduce the use of animal experiments, as proposed in the 3R principles formulated by Russell and Burch [43,44,45]. This makes methods that do not require in vitro tests (NTM) preferred, such as read-across approaches, along with other models involving statistical methods. These strategic approaches provide a quantitative structure–activity relationship (QSAR) [46].

Computer models are an important tool visualized in the current study. Using the available software tools Toxicity Estimation Software Tool (T.E.S.T.) version 5.1.2 [47] and VEGA version 1.2.3 [48], in silico predictive data for the properties and toxicity of our target molecules are presented. The VEGA tool has several advantages that can be applied in the development of new cosmetic products [48]. A widely used method for representing chemical structures in text format is SMILES (Simplified Molecular Input Line Entry System) [49,50,51]. In Table 4, the structures of our synthesized 2-arylbenzothiazole compounds are presented, along with well-known reference substances that are described in the literature—PBSA, thymol **3a**, and carvacrol **3b**, used for comparison of the conducted in silico analyses.

By applying in silico methods and using various models implemented in the software system VEGA—version 1.2.3 [48], the following properties of the compounds presented in Table 5 were successfully predicted—Mutagenicity, Acute Toxicity (LD_50_), Skin Irritation, Thyroid Receptor Alpha effect (TRα) and Beta effect (THRβ), Reproductive Toxicity. Regarding the mutagenicity of all tested compounds, a brief summary can be made as follows, the results obtained from the application of three different models classify the compounds **3a**,**b**, **6**, **7** and PBSA as non-mutagenic. Using the same mutagenicity models, BT-PhOH hybrid was also classified by two of them as non-mutagenic (Table 5).

Predictive results for toxicity were also obtained using the software tool T.E.S.T.—version 5.1.2. A machine learning method based on the nearest neighbour was successfully applied [47]. For all compounds, calculations were made to predict toxic properties against T. Pyriformis IGC_50_ (48 h) and Daphnia magna LC_50_ (48 h), with the results presented in Table 6, columns 4 and 5.

Using the same software, predictive results for Oral rat LD_50_ values were obtained. The predicted LD_50_ values for the tested compounds range from 38.30 mg/kg to 3192.7 mg/kg (Table 6).

The results obtained from the applied in silico platforms (Table 5 and Table 6), compounds BT-PhOH, **6** and **7** are classified as non-mutagenic, with lower toxicity, non-irritating to the skin, and without Thyroid receptor *alpha* effect (TR*α*) and *beta* effect (THR*β*). Nevertheless, to make a final conclusion about the properties of these compounds, additional in vitro and in vivo tests are necessary.

The predicted in silico results provided us with a good lead for continuing our research. Besides the design, synthesis, and prediction of the properties of the molecules through in silico platforms, we are also interested in investigations about their UV-filter activity. 

#### 2.2.2. In Vitro Determination of the Sun-Protection Factor (SPF)

The in vitro sun-protection factor (SPF) of some of the new synthetic analogues of 2-phenyl-1*H*-benzimidazole-5-sulfonic acid (PBSA), presented in Figure 2, was investigated. In the conducted in vitro studies, PBSA was chosen as the reference standard substance.

In this regard, a well-established spectrophotometric method was successfully ap-plied [52,53]. Table 7 shows the results for 0.0312 mM solution of compound **7** in 96% ethanol. The sun-protection factor (SPF) was evaluated and calculated using the Mansur equation [52] and EE × I values are constants determined by Sayre et al. (1979) [53]. Thus, the sun-protection factor at different concentrations was determined experimentally (Table 7). The results obtained for the remaining concentrations of 2-arylbenzazole compounds—BT-PhOH hybrid, 6, 7, and PBSA—were systematized and summarized.

The obtained results provide information about the relationship between the structure of the synthesized benzothiazole hybrid molecules and their UV-filter activity. The maximum concentration of PBSA approved by the Food and Drug Administration (FDA) is 148 mM. In commercial sunscreens, PBSA concentrations often range between 74 mM and 148 mM [24]. For the cosmetic industry, achieving a high SPF value at a low concentration of the tested compound is very valuable. We selected the following working concentrations for this investigation—from 1 mM to 0.005 mM (Table 8).

The UV spectra of the investigated compounds at the same concentration in the range of 270–370 nm are presented in Figure 3.

The spectra are presented in the range of 270–370 nm, as the substances were tested as UV filters in the UVB region between 290 nm and 320 nm. PBSA has proven sunscreen properties [24], and it shows a band covering this region with an absorption maximum at 305 nm and two at approximately 280 nm and 320 nm.

The results of the in silico predicted properties of the amidoalkylated compounds **4** and **5** (Table 5 and Table 6) and the lack of conjugated double bonds led to the hypothesis that the compounds might not be effective UV filters. Despite the expected results, we investigated the spectral properties of compound **4a** (Figure 3). The results confirmed our initial hypothesis that the spectrum characteristics are much influenced by the presence of an amide group and a BT ring.

We investigated the potential of compounds **6**, **7**, and BT-PhOH as UV filters in the considered region. In the UV spectrum of compound **7**, a broad absorption maximum is observed in the range of 280–315 nm, while its isomer, compound **6**, showed an absorption maximum at 314 nm with a shoulder at 290 nm in the range of 280–330 nm. The absorption of the investigated compounds in this region is due to the charge transfer from the electron-donating phenolic group at the para-position to the electron-accepting BT ring, resulting in an n → π* transition.

For **BT-PhOH hybrid**, the absorption maximum at 318 nm is due to the presence of a phenolic group at the para-position in the benzene ring, whose lone electron pair interacts with the π-electron cloud of the two delocalized aromatic systems. In contrast, compounds **6** and **7** exhibited a hypsochromic shift of the absorption maximum by 4 nm and 18 nm, respectively. The probable reason for this is the presence of electron-donating alkyl groups in these isomeric compounds, which affects the distribution of electron density and consequently increases the energy difference between the HOMO and LUMO levels, resulting in a shift of the absorption maximum to shorter wavelengths.

It was found that there is no absorption in the entire UV region in the UV spectrum of compound **4a**, and therefore measurements to calculate the SPF factor were not defined. The reason for this is the lack of conjugation throughout the molecule due to dearomatization in the benzothiazoline core.

It is known that the lack of spliced double bonds in the molecules significantly reduces the effectiveness of UV filters. The lack of such a structure limits the ability of molecules to absorb and neutralize UV rays, and such a characteristic is exceptional. Therefore, in our current research, our focus is on determining the sun-protection factor of aromatic compounds.

#### 2.2.3. Antimicrobial Activity Assay

Wild thyme, thymol, and carvacrol are used as antimicrobial agents in various foods and cosmetics [5,54]. This study aims to evaluate the antimicrobial sensitivity of standard test microorganisms, which are monitored and potentially could be found in cosmetic products, against a series of new synthetic 2-substituted benzothiazole compounds. 

The antibacterial activity of the synthesized hybrids **4a**,**b**, **5a**,**b**, **6**, and **7** was tested against a set of six microbial strains, *Escherichia coli* ATCC 25922, *Escherichia coli* ATCC 8739, *Staphylococcus aureus* ATCC 25923, *Staphylococcus aureus* ATCC 6538, *Pseudomonas aeruginosa* NBIMCC 1390, and *Candida albicans* NBIMCC 74, using the well-known agar diffusion method with wells (Table 9). The thyme oil, thymol, and carvacrol, purchased from commercial sources, were applied as reference substances. The specific conditions and consumables for the antimicrobial evaluation are clearly described in the Materials and Methods section (see Section 3.1.8).

The results presented in Table 9 show that the reference substances do not exhibit antibacterial activity against *Escherichia coli* ATCC 25922, *Escherichia coli* ATCC 8739, *Staphylococcus aureus* ATCC 25923, *Staphylococcus aureus* ATCC 6538, and *Pseudomonas aeruginosa* NBIMCC 1390 at the tested concentrations of 250 µg/mL, 500 µg/mL, and 1000 µg/mL. At these concentrations, the amount of each substance in the agar wells (6 mm) is 15 µg, 30 µg, and 60 µg, respectively. The concentration range of the tested substances was selected based on previous experience with essential oils and natural compounds. The reference substances exhibited antifungal activity against *Candida albicans* NBIMCC 74. For thyme oil and thymol, a more pronounced influence of the substance quantities on antifungal activity is observed. For thyme oil, the inhibition zones varied as follows: 9 mm, 10 mm, and 12 mm, when the substance amounts are 15 µg, 30 µg, and 60 µg, respectively. A similar tendency was observed for thymol, with inhibition zones ranging from 12 mm, 11 ± 0.71 mm, and 10 ± 0.71 mm. Carvacrol showed lower antifungal activity at the lowest substance amount of 15 µg, with an inhibition zone of 9 mm, while for the other two amounts, comparable inhibition zone values of 11 mm and 10 ± 0.71 mm for 60 µg and 30 µg, respectively, are observed. 

The data presented in Table 9 also show that synthetic analogues **6** and **7** exhibited activity similar to the reference substances. They did not inhibit bacterial growth but only exhibited antifungal activity. For compound **6**, antifungal activity is observed only at the highest amount of 60 µg, where the inhibition zone is 10 mm. Benzothiazolyl-2-carvacrol analogue **7** also showed weak antifungal activity, with comparable inhibition zones regardless of the substance amount—9 mm at 30 µg and 60 µg, and 8 ± 0.71 mm at 15 µg. This indicates that for compound **7**, changing the substance amount does not lead to a significant change in the analogue’s activity. A similar trend was observed for analogue **5a**. Its antifungal activity is not influenced by concentration, as the inhibition zones are of equal diameter—13 ± 0.071 mm—for all three tested concentrations. However, this analogue showed some antibacterial activity only against *Staphylococcus aureus* ATCC 25923 at amount of 60 µg, with an inhibition zone of 9 ± 0.71 mm. As a result of these studies, it can be concluded that both the reference substances and analogues **5a**, **6**, and **7** exhibited weak antimicrobial activity. 

Analogues **4a**, **4b**, and **5b** are characterized by high antibacterial and antifungal activity. They inhibited the growth of all test microorganisms. Analogue **4a** showed equal antimicrobial activity against both strains of *Escherichia coli*, with inhibition zones of 15 ± 0.71 mm at an amount of 60 µg. As the concentration of the analogue decreases, the inhibition zone decreases, but it is equal for the respective amounts of 15 µg and 30 µg—13 mm for both strains. Analogue **4a** exhibited different activity against the two strains of *Staphylococcus aureus*, being more effective against *Staphylococcus aureus* ATCC 6538, with inhibition zones of 16 ± 0.71 mm and 13 ± 0.71 mm, while for *Staphylococcus aureus* ATCC 25923, the inhibition zones are 11 ± 0.71 mm, 10 ± 0.71 mm, and 9 ± 0.71 mm for the three tested concentrations. This indicates that the sensitivity of *Staphylococcus aureus* representatives is different and strain-specific. Table 9 showed that the concentration of analogue **4a** did not affect its antifungal activity, as comparable inhibition zones of 12 mm and 12 ± 0.71 mm are observed for all three concentrations. Compound **4a** inhibited the growth of *Pseudomonas aeruginosa* NBIMCC 1390. For this pathogen, the substance amount slightly affects its antimicrobial activity, as indicated by the inhibition zones of 14 ± 0.71 mm for 60 µg and 13 mm for 15 µg and 30 µg. 

Analogue **4b** also exhibited antibacterial and antifungal activity. Notably, its antimicrobial activity against both strains of *Staphylococcus aureus* was equally high, confirmed by the equal inhibition zone sizes of 15 ± 0.71 mm for 60 µg and 14 mm for 15 µg and 30 µg of the substance. This suggests that this analogue may exhibit antimicrobial activity against different strains of *Staphylococcus aureus*. Analogue **4b** also showed comparable antimicrobial activity against both strains of *Escherichia coli*. The influence of the substance amount in the wells was more pronounced for the *Escherichia coli* ATCC 25922 strain, as the inhibition zone diameter decreased by about 2 mm with decreasing substance concentration. For this strain, the inhibition zones were 16 ± 0.71 mm, 14 ± 0.71 mm, and 12 mm for amounts of 60 µg, 30 µg, and 15 µg, respectively. For the *Escherichia coli* ATCC 8739 strain, the change in analogue concentration did not significantly affect antimicrobial activity, as indicated by the similar inhibition zone diameters of 15 ± 0.71 mm, 14 ± 0.71 mm, and 13 mm. The amount of **4b** in the agar wells against *Candida albicans* NBIMCC 74 did not affect its antifungal activity, as indicated by the comparable inhibition zones of 12 mm and 12 ± 0.71 mm (Table 9). For *Pseudomonas aeruginosa* NBIMCC 1390, a larger inhibition zone was observed at a substance amount of 60 µg—14 ± 0.71 mm. As the analogue amounts decreased to 30 µg and 15 µg, the inhibition zone diameter decreased by about 2 mm, with comparable zones of 12 ± 0.71 mm and 12 mm for the two concentrations.

The antimicrobial profile of **5b** was similar to compounds **4a** and **4b**. It should be noted that this analogue exhibited the highest antimicrobial activity against *Staphylococcus aureus* ATCC 25923, with inhibition zones of 24 ± 2.12 mm, 22 mm, and 20 mm. For the other strain, *Staphylococcus aureus* ATCC 6538, the inhibition zones were smaller, indicating that the sensitivity of different Staphylococcus aureus strains was strain-specific. A similar affinity was observed for the strains *Escherichia coli* ATCC 25922 and *Escherichia coli* ATCC 8739, where the inhibition zones were larger for Escherichia coli ATCC 8739 compared to *Escherichia coli* ATCC 25922 (Table 9). For the strains *Escherichia coli* ATCC 25922, *Escherichia coli* ATCC 8739, *Staphylococcus aureus* ATCC 6538, and *Staphylococcus aureus* ATCC 25923, a decrease in the diameter of the inhibition zones is observed, indicating that the concentration of the analogue rather affected its antimicrobial properties. A similar trend is detected for *Pseudomonas aeruginosa* NBIMCC 1390, where the inhibition zones decrease with the reduction in the substance amount in the agar wells—13 ± 0.71 mm, 12 ± 0.71 mm, and 10 mm. For *Candida albicans* NBIMCC 74, reducing the substance amount of the analogue from 60 µg to 30 µg led to a decrease in the inhibition zone from 13 ± 0.71 mm to 10 mm, and a further reduction to 15 µg resulted in maintaining the inhibition zone size at 10 mm.

The reference compounds did not exhibit activity, likely due to the testing conditions (Materials and Methods, Section 3.1.8) and the concentration used. The observed high antibacterial and antifungal activity of analogues **4a**, **4b**, and **5b** makes them valuable resources and serves as a good lead for further investigations and structure optimization.

#### 2.2.4. Radical Scavenging Activity Assay

The benzothiazole–terpenoid hybrids **4**–**7** contain a hydroxyphenyl fragment in their structures, suggesting potential radical scavenging activity, as established for similar structures by other authors [27]. This motivated the examination of the newly synthesized compounds for their in vitro radical scavenging activity. Gallic acid, quercetin, and rutin, known as good antioxidants, were compared with hybrids **4a**–**c**, **5a**–**c**, **6**, and **7** to evaluate their antioxidant activity potential against synthetic free radicals. Various methods are used in practice to determine antioxidant activity based on electron and hydrogen atom transfer. There is no single method to establish exact activity, as each substance reacts differently depending on the method’s principle. The choice of methods for investigating antioxidant activity, particularly their radical-scavenging ability, depends on the lipophilicity and structure of the compounds.

Based on our previous experience in determining the radical scavenging activity of polyphenolic and phenolic compounds [33,55], we chose to use relatively stable synthetic free radicals such as 2,2-diphenyl-1-picrylhydrazyl (DPPH method) and 2,2′-azinobis-3-ethyl-benzothiazoline sulfonic acid (ABTS method). To find a relationship between structure and activity, the activity of the synthesized 2-arylbenzothiazoles was investigated and compared with natural phenolic compounds. The analyses were performed under the same conditions for both methods (Table 10).

The radical scavenging activity of the DPPH method is quick and convenient for demonstrating the activity of potential antioxidant substances. As expected, the hybrid compounds of thymol **4a**,**b** and carvacrol **5a**,**b**, containing a BT fragment in their molecule, showed low radical scavenging activity > 1000 µM. The results can likely be attributed to the positioning of the free radical in the DPPH molecule and the structure of examined products.

Two of the newly synthesized hybrids, **4a** (133.70 ± 10 µM) and **4b** (157.50 ± 10 µM), demonstrated good radical scavenging activity by the ABTS method. The activity of the carvacrol hybrid molecules is lower, 300 ± 25 µM **5b**, compared to the thymol hybrids, 213 ± 20 µM **5a**. Compound **6** (421.30 ± 20 µM) has similar activity to the carvacrol hybrids (average 257 µM), while **7** has the lowest antioxidant activity (>2000 µM). The results for thymol hybrids by the ABTS method are close to those of natural thymol (92.50 ± 10 µM) and rutin (95.30 ± 4.50 µM) but significantly lower than those of quercetin (48.00 ± 4.40 µM) and gallic acid (37.70 ± 1.25 µM).

On the other hand, comparing the activity of the new hybrids **4**–**7** with 2-aryl benzothiazoles examined in our previous studies [26], the thymol hybrids are close to the compounds containing catechol (average 109 µM) and hydroquinone (average 120 µM). Additionally, the synthetic derivatives with one hydroxyl group showed extremely low radical scavenging activity (>1000 µM) by the ABTS method [55].

## 3. Materials and Methods

### 3.1. Chemistry

#### 3.1.1. General Information

All reagents, such as BT, thymol, carvacrol, PBSA, alkyl chloroformates, DDQ, etc., and organic solvents were purchased from commercial suppliers Sigma-Aldrich (Merck, Sofia, Bulgaria, EAD) and were used without further purification. Melting points were determined on a Boëtius PHMKO5 melting point meter (Carl Zeiss Jena, Germany) and are uncorrected. IR spectra were measured on VERTEX 70 FT-IR spectrometer (Bruker Optics, Ettlingen, Germany). HRMS spectral measurements were performed on a Bruker MS spectrometer. NMR spectra were measured on Bruker Avance AV600 spectrometer (Bruker, Billerica, MA, USA) at BAS-IOCCP—Sofia in DMSO-d_6_ as solvent. To average out the rotamers reaching adequate assignment of peaks and structure determination, the spectra were measured at 80 °C. Chemical shifts (δ, ppm) are referenced to TMS. The values of coupling constants (J) are calculated in Hz. Thin-layer chromatography (TLC) was performed on precoated 0.2 mm Merck silica gel 60 plates. Silica gel was used for column chromatographic separation.

#### 3.1.2. General Procedure for the Synthesis of Benzothiazole–Monoterpenoid Hybrids **4a**–**c** and **5a**–**c**

To 2 mmol of benzothiazole dissolved in 8 mL of dichloromethane, an excess of the used ethyl or methyl chloroformate (2.4 mmol) and the corresponding nucleophilic reagent (thymol or carvacrol) were successively added. The reaction mixture was stirred at room temperature for a specified period, as described in Table 1. The completion of the reactions was monitored by TLC. The resulting crude product was purified by column chromatography on silica gel as stationary phase and eluent mixtures of petroleum/diethyl ether in various ratios. The products were isolated as white crystals. 

Compound **4a** (Ethyl-2-(4-hydroxy-5-isopropyl-2-methylphenyl)benzo[d]thiazole-3(2H)-carboxylate): isolated by column chromatography on silica gel with mixtures of petroleum/diethyl ether (8:1), yield: 96%, m.p. = 140–142 °C, MW = 357.47 g/mol.

^1^H-NMR (600 MHz, 80 °C, DMSO-d_6_, *δ* ppm, *J* Hz): 0.92 (d, *J* = 7.0, 3H, -CH(CH_3_)_2_-thymol), 0.99 (d, *J* = 7.0, 3H, -CH(CH_3_)_2_-thymol), 1.12 (t, *J* = 7.0, 3H, -COOCH_2_CH_3_), 2.23 (s, 3H, -CH_3_-thymol), 3.02–3.06 (m, 1H, -CH(CH_3_)_2_-thymol), 4.14–4.17 (m, 2H, -COOCH_2_CH_3_), 6.60 (s, 1H, -CH-thymol), 6.82 (s, 1H, -CH-thymol), 6.85 (s, 1H, -*CH), 7.03 (t, *J* = 7.6, 1H, -CH-benzothiazole), 7.17 (t, *J* = 8.2, 1H, -CH-benzothiazole), 7.21 (d, *J* = 7.6, 1H, -CH-benzothiazole), 7.82 (d, *J* = 8.2, 1H, -CH-benzothiazole), 8.95 (brs, 1H, OH-thymol).

^13^C-NMR (150 MHz, 80 °C, DMSO-d_6_, *δ* ppm, *J* Hz): 14.5, 18.7, 22.7, 22.8, 26.6, 62.4, 64.5, 116.8, 117.8, 122.2, 122.9, 124.6, 125.8, 128.4, 131.1, 132.0, 132.4, 139.0, 152.7, 154.6.

IR (KBr, cm^−1^): 3401, 3064, 2962, 1673, 1580, 1474, 1397, 1330, 1266, 1040, 854, 744.

HRMS m/z (ESI): calcd for C_20_H_22_NO_3_S^−^ [M-H]^−^ 356.1326, found 356.1324.

Compound **4b** (Methyl-2-(4-hydroxy-5-isopropyl-2-methylphenyl)benzo[*d*]thiazole-3(2*H*)-carboxylate): isolated by column chromatography on silica gel with mixtures of petroleum/diethyl ether (8:1), yield: 90 %, m.p. = 131–133 °C, MW = 343.44 g/mol.

^1^H-NMR (600 MHz, 80 °C, DMSO-d_6_, *δ* ppm, *J* Hz): 0.91 (d, *J* = 7.0, 3H, -CH(CH_3_)_2_-thymol), 1.00 (d, *J* = 7.0, 3H, -CH(CH_3_)_2_-thymol), 2.24 (s, 3H, -CH_3_-thymol), 3.01–3.06 (m, 1H, -CH(CH_3_)_2_-thymol), 3.71 (s, 3H, -COOCH_3_), 6.60 (s, 1H, -CH-thymol), 6.81 (s, 1H, -CH-Thymol), 6.85 (s, 1H, -*CH), 7.04 (t, *J* = 7.6, 1H, -CH-benzothiazole), 7.18 (t, *J* = 8.2, 1H, -CH-benzothiazole), 7.21 (d, *J* = 7.6, 1H, -CH-benzothiazole), 7.81 (d, *J* = 8.2, 1H, -CH-benzothiazole), 8.96 (s, 1H, OH-thymol).

^13^C-NMR (150 MHz, 80 °C, DMSO-d_6_, *δ* ppm, *J* Hz): 18.7, 22.7, 22.8, 26.6, 53.6, 64.5, 117.0, 117.9, 122.1, 123.0, 124.8, 125.8, 128.5, 131.0, 132.0, 138.9, 139.0, 153.3, 154.6. 

IR (KBr, cm^−1^): 3391, 3073, 2961, 1683, 1580, 1474, 1438, 1340, 1242, 1041, 856, 738.

HRMS m/z (ESI): calcd for C_19_H_20_NO_3_S^−^ [M-H]^−^ 342.1169, found 342.1165.

Compound **4c** (2,2,2-trichloroethyl-2-(4-hydroxy-5-isopropyl-2-methylphenyl) benzo[*d*]thiazole-3(2*H*)-carboxylate): isolated by column chromatography on silica gel with mixtures of petroleum/diethyl ether (2:1), yield: 82%, m.p. = 145–147 °C, MW = 460.79 g/mol.

^1^H-NMR: (600 MHz, 80 °C DMSO-d_6_, *δ* ppm, *J* Hz): 0.90 (d, *J* = 6.46, 3H, -CH(CH_3_)_2_-thymol), 0.99 (d, *J* = 7.04, 3H, -CH(CH_3_)_2_-thymol), 2.24 (s, 3H, -CH_3_-thymol), 3.02—3.05 (m, 1H, -CH(CH_3_)_2_-thymol), 4.95 (d, 2H, -COOCH_2_Cl), 6.60 (s, 1H, -CH-thymol), 6.82 (s, 1H, -CH-thymol), 6.93 (s, 1H, -*CH), 7.10 (t, *J* = 7.63, 1H, -CH-benzothiazole), 7.22 (t, *J* = 8.22, 1.17, 1H, -CH-benzothiazole), 7.26 (d, *J* = 7.63, 1.17, 1H, -CH-benzothiazole), 7.91 (d, *J* = 8.22, 1H, -CH-benzothiazole), 8.94 (s, 1H, OH-thymol).

^13^C-NMR: (150 MHz, 80 °C, DMSO-d_6_, δ ppm, *J* Hz) 18.75, 22.55, 22.81, 26.70, 64.59, 75.29, 95.73, 117.09, 117.93, 121.83, 123.50, 123.58, 125.46, 125.94, 130.58, 131.91, 132.45, 154.72.

IR (KBr, cm^−1^): 3454, 3072, 2964, 1710, 1579, 1514, 1471, 1265, 824, 754.

HRMS m/z (ESI): calcd for C_20_H_19_Cl_3_NO_3_S^−^ [M-H]^−^ 458.0157, found 458.0160.

Compound **5a** (Ethyl-2-(4-hydroxy-2-isopropyl-5-methylphenyl)benzo[*d*]thiazole-3(2*H*)-carboxylate): isolated by column chromatography on silica gel with mixtures of petroleum/diethyl ether (8:1, increasing polarity to 4:1), yield: 93%, m.p. = 192–193 °C, MW = 357.47 g/mol.

^1^H-NMR: (600 MHz, 80 °C, DMSO-d_6_, *δ* ppm, *J* Hz): 1.12 (t, *J* = 7.0, 3H, -COOCH_2_CH_3_), 1.20 (d, *J* = 6.5, 3H, -CH(CH_3_)_2_-carvacrol), 1.23 (d, *J* = 7.0, 3H, -CH(CH_3_)_2_-carvacrol), 1.95 (s, 3H, -CH_3_-carvacrol), 3.08–3.12 (m, 1H, -CH(CH_3_)_2_-carvacrol), 4.10–4.18 (m, 2H, -COOCH_2_CH_3_), 6.72 (s, 1H, -CH-carvacrol), 6.74 (s, 1H, -CH-carvacrol), 6.97 (s, 1H, -*CH), 7.03 (t, *J* = 7.6, 1H, -CH-benzothiazole), 7.17 (t, *J* = 8.2, 1H, -CH-benzothiazole), 7.19 (d, *J* = 7.6, 1H, -CH-benzothiazole), 7.84 (d, *J* = 8.2, 1H, -CH-benzothiazole), 8.93 (s, 1H, OH-carvacrol).

^13^C-NMR: (150 MHz, 80 °C, DMSO-d_6_, *δ* ppm, *J* Hz): 14.4, 16.0, 23.9, 24.1, 28.5, 62.4, 63.3, 112.5, 116.9, 122.1, 122.8, 124.7, 125.7, 126.3, 128.2, 130.4, 139.0, 143.1, 152.7, 156.0. 

IR (KBr, cm^−1^): 3434, 3056, 2958, 1704, 1575, 1470, 1348, 1268, 1151, 1027, 825, 743.

HRMS m/z (ESI): calcd for C_20_H_22_NO_3_S^−^ [M-H]^−^ 356.1326, found 356.1329.

Compound **5b** (methyl-2-(4-hydroxy-2-isopropyl-5-methylphenyl)benzo[*d*]thiazole-3(2*H*)-carboxylate): isolated by column chromatography on silica gel with mixtures of petroleum/diethyl ether (8:1), yield: 84%, m.p. = 142–144 °C, MW = 343.44 g/mol.

^1^H-NMR: (600 MHz, 80 °C, DMSO-d_6_, *δ* ppm, *J* Hz): 1.21 (d, *J* = 7.0, 3H, -CH(CH_3_)_2_-carvacrol), 1.23 (d, *J* = 6.5, 3H, -CH(CH_3_)_2_-carvacrol), 1.94 (s, 3H, -CH_3_-carvacrol), 3.07–3.12 (m, 1H, -CH(CH_3_)_2_-carvacrol), 3.70 (s, 3H, -COOCH_3_), 6.71 (s, 1H, -CH-carvacrol), 6.74 (s, 1H, -CH-carvacrol), 6.97 (s, 1H, -*CH), 7.04 (t, *J* = 7.6, 1H, -CH-benzothiazole), 7.17 (t, *J* = 8.2, 1H, -CH-benzothiazole), 7.20 (d, *J* = 7.6, 1H, -CH-benzothiazole), 7.81 (d, *J* = 8.2, 1H, -CH-benzothiazole), 8.94 (s, 1H, OH-carvacrol).

^13^C-NMR: (150 MHz, 80 °C, DMSO-d_6_, *δ* ppm, *J* Hz): 16.0, 23.8, 24.3, 28.5, 53.5, 63.4, 112.5, 117.0, 122.1, 122.9, 124.8, 125.8, 126.2, 128.3, 130.3, 138.9, 143.2, 153.3, 156.0. 

IR (KBr, cm^−1^): 3361, 3016, 2960, 1680, 1594, 1507, 1466, 1273, 1024, 885, 754.

HRMS m/z (ESI): calcd for C_19_H_20_NO_3_S^−^ [M-H]^−^ 342.1169, found 342.1172.

Compound **5c** 2,2,2-trichloroethyl-2-(4-hydroxy-2-isopropyl-5-methylphenyl)-benzo[*d*]thiazole-3(2*H*)-carboxylate isolated by column chromatography on silica gel with mixtures of petroleum/diethyl ether (2:1), yield: 86%, m.p. = 131–133 °C, MW = 460.79 g/mol. 

^1^H-NMR: (600 MHz, 80 °C, DMSO-d_6_, *δ* ppm, *J* Hz): 1.22 (dd, *J* = 8.22, 7.04 Hz, 6H, -CH(CH_3_)_2_-carvacrol), 1.94 (s, 3H, -CH_3_-carvacrol), 3.07–3.10 (m, 1H, -CH(CH_3_)_2_-carvacrol), 4.96 (s, 2H, -COOCH_2_Cl), 6.72 (s, 1H, -CH-carvacrol), 6.74 (s, 1H, -CH-carvacrol), 7.04 (s, 1H, -*CH), 7.07–7.12 (m, 1H, -CH-benzothiazole), 7.18–7.23 (m, 1H, -CH-benzothiazole), 7.24–7.27 (m, 1H, -CH-benzothiazole), 7.92 (d, *J* = 7.63 Hz, 1H, -CH-benzothiazole), 8.93 (s, 1H, OH-carvacrol). 

^13^C-NMR: (151 MHz, DMSO-d_6_) δ ppm 16.0, 24.0, 24.2, 28.6, 63.8, 75.3, 95.7, 112.8, 117.4, 122.0, 123.2, 125.5, 126.0, 126.2, 128.5, 129.5, 138.2, 143.1, 143.3, 143.3, 156.1; 

IR (KBr, cm^−1^): 3461, 3062, 2960, 1704, 1578, 1507, 1472, 1275, 1030, 843, 758;

HRMS m/z (ESI): calcd for C_20_H_19_Cl_3_NO_3_S^−^ [M-H]^−^ 458.0157, found 458.0153.

#### 3.1.3. General Procedure for the Oxidation of Thymol and Carvacrol Containing Benzothiazolines **4a**–**c** and **5a**–**c** Toward Aromatized Products **6** and **7**

To the corresponding compound **4a**–**c**, **5a**–**c** (0.5 mmol), dissolved in CH_3_CN (10 mL/mmol), the oxidant DDQ (1 mmol) was added. The reaction mixture was stirred with a magnetic stirrer at room temperature for 2 to 3 h (Table 3). After the completion of the reaction (monitored by TLC), the product precipitation was observed. The precipitate was filtered through a short filter, and the filtrate was extracted in a separatory funnel with water. The organic layer was dried with anhydrous sodium sulfate, and the solvent was evaporated. White crystalline products were isolated by recrystallization from petroleum ether.

Compound **6** (2-(4-hydroxy-5-isopropyl-2-methylphenyl)benzo[*d*]thiazole): isolated by column chromatography on silica gel with mixtures of petroleum/diethyl ether (2:1), yield: 70% (from **4a**); 71% (from **4b**) and 56% (from **4c**), m.p. = 195–197 °C, MW = 283.39 g/mol.

^1^H-NMR: (600 MHz, 80 °C, DMSO-d_6_, *δ* ppm, *J* Hz): 1.17 (d, *J* = 7.0, 6H, -CH(CH_3_)_2_-thymol), 2.50 (s, 3H, -CH_3_-thymol), 3.16–3.21 (m, 1H, -CH(CH_3_)_2_-thymol), 6.75 (s, 1H, -CH-thymol), 7.34 (t, *J* = 8.2, 1H, -CH-benzothiazole), 7.43 (t, *J* = 8.2, 1H, -CH-benzothiazole), 7.52 (s, 1H, -CH-thymol), 7.93 (d, *J* = 8.2, 1H, -CH-benzothiazole), 7.98 (d, *J* = 7.6, 1H, -CH-benzothiazole), 9.52 (s, 1H, OH-thymol).

^13^C-NMR: (150 MHz, 80 °C, DMSO-d_6_, δ ppm, *J* Hz): 21.3, 22.8, 26.9, 118.7, 122.0, 123.0, 124.0, 125.2, 126.6, 129.1, 133.2, 135.0, 135.9, 154.0, 157.0, 168.4. 

IR (KBr, cm^−1^): 3120, 2957, 1613, 1578, 1495, 1240, 1053, 867, 754.

HRMS m/z (ESI): calcd for C_17_H_16_NOS^−^ [M-H]^−^ 282.0958, found 282.0963. 

Compound **7** (2-(4-hydroxy-2-isopropyl-5-methylphenyl)benzo[*d*]thiazole): isolated by column chromatography on silica gel with mixtures of petroleum/diethyl ether (2:1), yield: 92% (from **5a**); 82% (from **5b**) and 41% (from **5c**), m.p. = 237–238 °C, MW = 283.39 g/mol.

^1^H-NMR: (600 MHz, 80 °C, DMSO-d_6_, *δ* ppm, *J* Hz): 1.19 (d, *J* = 7.0, 6H, -CH(CH_3_)_2_-carvacrol), 2.18 (s, 3H, -CH_3_-carvacrol), 3.79–3.83 (m, 1H, -CH(CH_3_)_2_-carvacrol), 6.95 (s, 1H, -CH-carvacrol), 7.37 (s, 1H, -CH-carvacrol), 7.42 (t, *J* = 8.2, 1H, -CH-benzothiazole), 7.51 (t, *J* = 8.2, 1H, -CH-benzothiazole), 7.99 (d, *J* = 8.2, 1H, -CH-benzothiazole), 8.05 (d, *J* = 7.6, 1H, -CH-benzothiazole), 9.43 (s, 1H, OH-carvacrol).

^13^C-NMR: (150 MHz, 80 °C, DMSO-d_6_, *δ* ppm, *J* Hz): 15.6, 24.2, 28.9, 113.0, 122.1, 123.1, 125.3, 126.6, 133.5, 135.5, 147.1, 154.0, 158.1, 168.3.

IR (KBr, cm^−1^): 3196, 2970, 1611, 1574, 1454, 1266, 1053, 891, 761. 

HRMS m/z (ESI): calcd for C_17_H_16_NOS^−^ [M-H]^−^ 282.0958, found 282.0955.

4-(benzo[d]thiazol-2-yl)phenol (BT-PhOH hybrid):

This compound is obtained and purified following our previously reported research [26].

#### 3.1.4. General Procedure for the Synthesis of Compounds **4a**,**b** and **5a**,**b** from Essential Oils of Thyme (*Thymus vulgaris*) and Oregano (*Origanum vulgare*) 

The α-amidoalkylation reactions were carried out by direct mixing of 2 mmol of benzothiazole dissolved in 8 mL of dry dichloromethane followed by the adding of an excess of the corresponding ethyl- or methyl chloroformate (3.5 mmol) at room temperature. After 20 min, 1.2 g thyme oil (4.3 mmol thymol) or 0.488 g oregano oil (2.5 mmol carvacrol) dissolved in 2 mL of dry dichloromethane was successively added dropwise. Then, the reaction mixture is stirred at room temperature for 24 h (monitored by TLC). The resulting crude product is purified by column chromatography on silica gel as stationary phase and eluents mixtures of petroleum/diethyl ether in various ratios. The pure products are isolated as white crystals with yields of 55% for **4a**, 62% for **4b**, 55% for **5a**, and 60% for **5b**, consequently.

#### 3.1.5. GC-MS/MS Analysis of Thyme and Oregano Essential Oils

The analyzed EOs were purchased from the commercial suppliers. Their chemical composition was investigated using GC-MS/MS analysis by a recently published method [37]. The published procedure, method and chromatographic conditions are used from the corresponding article [37]. The research examination was conducted on a TSQ 9000 chromatographic system (Thermo Fisher Scientific, Waltham, MA, USA), under an EI ionization mode at 70 eV and programmable temperature vaporizing (PTV) injector. 

#### 3.1.6. In Silico QSAR Analysis

In the present in silico prediction for selected target properties, we applied computational calculations using two open-access software systems—T.E.S.T. version 5.1.2 and VEGA version 1.2.3. We provided information on the properties of our newly synthesized 2-arylbenzothiazole compounds and the reference standard compounds (Table 5 and Table 6). 

The T.E.S.T. software tool (version 5.1.2) for toxicity assessment was developed by the United States Environmental Protection Agency (EPA, Washington, DC, USA) [47]. For the compounds presented, the machine learning method based on the nearest neighbour was applied. The other platform used is VEGA, a Java-based system developed by the Italian Istituto di Ricerche Farmacologiche Mario Negri. The VEGA software (version 1.2.3) has well-developed QSAR models for predicting toxic, ecotoxic, environmental, physicochemical, and toxicokinetic properties of chemical substances [48]. Using VEGA, the following properties of the target molecules were predicted: Mutagenicity, Acute Toxicity (LD_50_), Skin Irritation, Thyroid Receptor Alpha effect (TRα), Thyroid Receptor Beta effect (THRβ), and Reproductive Toxicity. The following models were applied for prediction of various effects, such as Mutagenicity (Ames test) model (CAESAR) 2.1.14, Mutagenicity (Ames test) model (ISS) 1.0.3, Mutagenicity (Ames test) model (SarPy-IRFMN) 1.0.8, Skin Irritation (CONCERT/Kode) 1.0.0, (CONCERT/Coral) 1.0.0, Acute Toxicity (LD50) model (KNN) 1.0.0, Thyroid Receptor Alpha and Beta effect (NRMEA) 1.0.1, Reproductive Toxicity library (PG) 1.1.2.

Both platforms, T.E.S.T. [47] and VEGA [48], provide an assessment of the toxic properties and environmental effects that the synthetic hybrid molecules might exert.

#### 3.1.7. Method for Determining the SPF Factor

An approved and standardized method for in vitro evaluation of UVA protection is described in ISO 24443:2021 [57]. However, there is still no approved in vitro method for evaluating UVB protection. The method proposed by Diffey–Robson for in vitro evaluation of the SPF parameter in 1989 is still successfully applicable today [58]. In our current study, the sun-protection factor (SPF) was evaluated and calculated using the Mansur equation, as calculated using Equation (1) [52,53]:
(1)$$SPF\;spectrophotometric=CF\;\times \overset{320}{\underset{290}{\sum }}EE\left(\lambda \right)\;\times \;I\;\left(\lambda \right)\;\times \;Abs\;\left(\lambda \right)$$
where EE is the erythemal effect spectrum, I is the solar intensity spectrum, Abs is the absorption of the sunscreen product or substance, CF is the correction factor (10), and λ is the wavelength (nm) [52]. The values of EE × I (Table 11) are constant published by Sayre et al. [53]. These normalized EE × I values are used for SPF calculation. 

In the in vitro evaluation of the SPF parameter, the tested compounds (**BT-PhOH hybrid**, **6**, **7**, and **PBSA**) were weighed and dissolved in a 50 mL volumetric flask with 96% ethanol as the solvent. From the obtained stock solution with a concentration of 1 mM, working solutions with the following concentrations were prepared by dilution: 0.5 mM, 0.25 mM, 0.125 mM, 0.0625 mM, 0.0312 mM, 0.01 mM, and 0.005 mM. The spectrophotometer was calibrated against the used solvent, 96% ethanol, for each wavelength. Each of the obtained solutions with different concentrations was tested separately in three consecutive repetitions. UV-VIS spectra were measured every 5 nm in the range of 290–320 nm. The Erythema effect (EE) and the solar intensity spectrum (I) are constants for each wavelength in the UVB spectrum, as shown in Table 11. The resulting absorption for each wavelength is multiplied by the corresponding constant EE × I. The resulting values for each repetition are added up, and the resulting sum is multiplied by the correction factor (CF), which is 10. Detailed calculations for compound **7** are presented in Table 7. 

Mansour’s equation was adapted for real-life conditions of sun exposure. This spectrophotometric method is quick, sensitive, selective, and suitable for the in vitro determination of SPF values of various sunscreen products, extracts, and synthetic substances.

For the purpose of the analysis, a dual-beam spectrophotometer Agilent Cary 60 UV/VIS (Agilent Technologies, Santa Clara, CA, USA), with an operational spectral range of 190–800 nm, was used.

#### 3.1.8. Method for Determination of Antimicrobial Activity

Test microorganisms: 

*Escherichia coli* ATCC 25922, *Escherichia coli* ATCC 8739, *Staphylococcus aureus* ATCC 25923, *Staphylococcus aureus* ATCC 6538, *Pseudomonas aeruginosa* NBIMCC 1390, and *Candida albicans* NBIMCC 74 from the collection of the Department of Microbiology and Biotechnology, University of Food Technologies, Plovdiv, Bulgaria. 

Nutrient medium: 

LBG-agar. Composition (g/dm^3^): tryptone—10; yeast extract—5; NaCl—10; glucose—10; agar—20. pH = 7.5. Sterilization—121 °C for 20 min. 

The antimicrobial activity was studied by the agar-diffusion method [59]. Suspensions with an active cell concentration of 108 cfu/cm^3^ were prepared from *Escherichia coli* ATCC 25922, *Escherichia coli* ATCC 8739, *Staphylococcus aureus* ATCC 25923, *Staphylococcus aureus* ATCC 6538, and *Pseudomonas aeruginosa* NBIMCC 1390 after 24-h cultivation in a thermostat at 37 °C on LBG-agar medium, using McFarland standard 0.5. The suspension of *Candida albicans* NBIMCC 74 had an active cell concentration of 107 cfu/cm^3^. The obtained suspensions were used to inoculate Petri dishes with LBG-agar medium, and after the agar solidified, wells (6 mm) were prepared. In each well, 60 µL of the tested substance was added, and the Petri dishes with the test microorganisms were incubated at the optimal temperature for the respective test microorganisms for 24 to 48 h. Antimicrobial activities were determined by measuring the inhibition zones in mm. The results of these studies are presented in Table 9.

#### 3.1.9. Radical Scavenging Activity Assay

##### DPPH Free Radical Scavenging Assay 

The DPPH free radical scavenging activities were measured as previously reported by Docheva et al. [56]: 0.12 mM DPPH was dissolved in methanol. The absorbance change was measured at 515 nm on a UV-Vis spectrophotometer within 30 min. The total DPPH radical scavenging activity within 30 min was measured in triplicate in the absence of light. The blank sample was prepared as above by replacing the test sample with equivalent methanol. The radical scavenging activity (RSA%) was calculated. IC_50_ value determined the effective concentration at which 50% of DPPH radicals were scavenged, and it was obtained by interpolation from linear regression analysis. A lower IC_50_ value indicates a higher antioxidant activity.

##### ABTS Free Radical Scavenging Assay 

The ABTS free radical was prepared by the method of Re et al. [60], with some modification. ABTS^·+^ was produced by 7 mM ABTS and 2.45 mM K_2_S_2_O_8_ dissolved in deionized H_2_O (the mixture stayed in the dark at room temperature for 12–16 h before use). A mixture of the reagents as 1:1 (*v*/*v*) ABTS^·+^ solution was diluted with methanol to an absorbance of 0.70 ± 0.02 at 734 nm. The ABTS radical scavenging activity within 10 min was measured in triplicate in the absence of light at room temperature. The percentage inhibition (%) of radical scavenging activity was calculated according to the corresponding equation of the method.

The stock solution of all compounds was prepared in a concentration of 1 mg/mL. The working solutions were prepared by dissolving aliquot parts of the stock solution with methanol. The final concentration of IC_50_ was calculated according to the dilution factor. 

#### 3.1.10. STATISTICS

The data obtained were expressed as the mean ± standard error of the mean (SEM). All statistical analyses were performed using the specialized software, SPSS, version 16.0 (SPSS Inc., Chicago, IL, USA).

## 4. Conclusions

The synthesis of multifunctional hybrid molecules is gaining popularity in the application of EOs in organic synthesis. This study was commenced to investigate and assess the synthesis and spectral characterization of 2-substituted benzothiazole derivatives, incorporating natural monoterpenoids, such as thymol and carvacrol, found in Thyme’s EO. The scope of the amidoalkylation reaction of phenolic compounds with *N*-acyliminium reagents derived from benzothiazole and alkyl chloroformates was extended to obtain eight benzothiazole–monoterpenoid hybrids **4**–**7** with high yields, ranging from 82% to 96%. Given the drawbacks of classical condensation reactions, the use of the amidoalkylation reaction offers significant advantages for the synthesis of hybrid molecules. This approach combines the accessibility of starting reagents with mild reaction conditions. This method enables the synthesis of multifunctional compounds, making it a valuable tool in organic synthesis for the design of new bioactive substances. The possibility of the synthesis of target substances using essential oils’ thymol and carvacrol chemotypes was successfully demonstrated. 

It is established that the *α*-amidoalkylation reactions proceeded regioselectively at the *para*-position relative to the hydroxyl group in thymol and carvacrol for the synthesis of compounds **4a**–**c** and **5a**–**c**. The oxidation reaction conditions for their rearomatization were established. Products **6** and **7** were obtained using DDQ in acetonitrile, with the highest yields being 71% and 92% after filtration and recrystallization. The structure of all new molecular hybrids was confirmed by spectral methods. 

The spectral properties of the selected compounds **6** and **7** were investigated, and the sun-protection factor (SPF) was determined using an established spectrophotometric method based on the Mansur equation. The obtained results showed SPF values were close to those of the commercial UVB filter PBSA at concentrations from 1 mM to 0.25 mM. The antimicrobial susceptibility was examined for a set of six microbial strains—*Escherichia coli* ATCC 25922, *Escherichia coli* ATCC 8739, *Staphylococcus aureus* ATCC 25923, *Staphylococcus aureus* ATCC 6538, *Pseudomonas aeruginosa* NBIMCC 1390, and *Candida albicans* NBIMCC 74—using the agar diffusion method with wells. The obtained results were compared with those of the reference substances (thyme oil, thymol, and carvacrol). The high antibacterial and antifungal activity of compounds **4a**, **4b**, and **5b** makes them potential antimicrobial agents. Compound **5b** showed the highest activity against *Staphylococcus aureus* ATCC 25923 at a tested amount of 0.044 μmol and measured inhibition zone of 20 mm. The radical scavenging activity was evaluated using DPPH and ABTS assays. The highest radical scavenging activity using the ABTS method exhibited thymol hybrids **4a** (IC_50_—133.70 ± 10 µM) and **4b** (IC_50_—157.50 ± 10 µM). The results of the in silico analysis indicated that compounds **6** and **7** are the most promising for further in vitro and in vivo testing. As a result of the conducted studies, we consider that the new hybrid compounds, compounds **6** and **7**, could be promising hit molecules with potential application in cosmetic products.

## Data Availability

The data presented in this study are available in this article and Appendix A.

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
