# Peer review of "New Benzothiazole–Monoterpenoid Hybrids as Multifunctional Molecules with Potential Applications in Cosmetics"

_molecules, 2025, doi:10.3390/molecules30030636_

Round 1
Reviewer 1 Report
Comments and Suggestions for Authors
The manuscript has a certain scientific novelty, it is written in simple understandable language and can be published after appropriate corrections.
1. In my opinion, it is inappropriate to include compound numbers in the abstract. I suggest replacing them with the specific names of the compounds.
2. Line 87: I believe references 10 and 20 do not correspond to the text: Although melanin serves to protect the skin from harmful UV rays, overproduction can lead to skin defects such as pigmentation changes, freckles, etc.
3. Reference 21, in my opinion, is outdated. For example, there are newer studies addressing the issue of skin cancer caused by ultraviolet rays.
4. Lines 125–132: The corresponding references should be added to this section of the manuscript.
5. Reaction products are missing from Scheme 1.
6. In the experimental section, nothing is mentioned about compounds 5 and 6. Were they purchased, or were they obtained according to the described methodology?
7. Figure 1 depicts the structures of benzothiazoles with aryl substituents that have already been described in the literature. In my opinion, the numbering of benzothiazoles in Figure 1 should be removed, as their specific names are already provided. I suggest starting the numbering of the compounds described in the authors’ work from 1.
8. I propose adding the synthesis method for compounds 7a and 7b using essential oil to the experimental section.
9. The manuscript text does not include a description of the preparation of compounds 9 and 10.
10. I also suggest removing the numbering in Figure 3.
11. What is the rationale behind the selected concentration range for determining the sun protection factor (SPF) in vitro? What solvent was used to prepare the concentrations of the studied compounds?
12. I propose adding references to the synthesis methods for compounds 2 and 11 to the experimental section since their synthesis is mentioned in the text but is not described in the results or the experimental section.
13. What solvent was used to prepare the concentrations of the studied compounds in the investigation of their antimicrobial properties? Was the stability of the studied compounds in the solvent used for antimicrobial activity studies evaluated?
14. The authors of the manuscript should unify the description of the 1H^1 \mathrm{H}1H-NMR spectra. When describing the spectra of the obtained compounds, specifically doublet, triplet, and multiplet signals, the authors sometimes use the range of signal values and sometimes use the mean value. For example, for compound 7b, the signal is described as 7.21 (d, J=7.6J = 7.6J=7.6, 1H, -CH-Benzothiazole), while for compound 7c, it is 4.90–4.99 (d, 2H, -COOCH2_22Cl).
15. In the experimental section, the 1H^1 \mathrm{H}1H-NMR spectra should list the proton signals separated by commas (compounds 7c and 8c).
16. There are errors in the compound names: (4-(benzo[d]thiazol-2-yl)-2-isopropyl-5-methylphenol) and (3-(benzo[d]thiazol-2-yl)-5-isopropyl-2-methylphenol), specifically in the term "isopropyl." Corrections are necessary.
Author Response
Response to Reviewer 1 Comments
Dear Reviewer, thank you for your effort to provide timely review and scientific decision on our manuscript.
Reviewer 1 Comments
The manuscript has a certain scientific novelty, it is written in simple understandable language and can be published after appropriate corrections.
Author’s comments: Thank you for the positive point of view and scientific discussion on our work.
- In my opinion, it is inappropriate to include compound numbers in the abstract. I suggest replacing them with the specific names of the compounds.
Author’s response: The compound numbers are correctly replaced with the specific names of the active compounds. The abstract is corrected in the revised version of the manuscript.
- Line 87: I believe references 10 and 20 do not correspond to the text: Although melanin serves to protect the skin from harmful UV rays, overproduction can lead to skin defects such as pigmentation changes, freckles, etc.
Author’s response: Thank you for the recommendation. The sentence has been removed and replaced with a new and up-to-date references [20,21]
- Reference 21, in my opinion, is outdated. For example, there are newer studies addressing the issue of skin cancer caused by ultraviolet rays.
Author’s response: In connection with the context of the previous remark, the text was continued and literature [21] was replaced with newer study and current literature connected to the subject of the article.
- Lines 125–132: The corresponding references should be added to this section of the manuscript.
Author’s response: The corresponding references are correctly introduced. The sentence was modified and relevant citations were added to make it more apparent to readers.
- Reaction products are missing from Scheme 1.
Author’s response: Thank you for the constructive remark. The reaction products are added in the revised manuscript text.
- In the experimental section, nothing is mentioned about compounds 5 and 6. Were they purchased, or were they obtained according to the described methodology?
Author’s response: Thank you for the correct observation. The information is added in the revised version of the manuscript (Experimental section, 3.1.1. General information).
- Figure 1 depicts the structures of benzothiazoles with aryl substituents that have already been described in the literature. In my opinion, the numbering of benzothiazoles in Figure 1 should be removed, as their specific names are already provided. I suggest starting the numbering of the compounds described in the authors’ work from 1.
Author’s response: It is corrected in the revised version of the manuscript. The numbers of the compounds are reformulated.
- I propose adding the synthesis method for compounds 7a and 7b using essential oil to the experimental section.
Author’s response: The method procedure for the obtaining of compounds 7a and 7b using essential oil is added in experimental section (3. Materials and Methods; 3.1.4. General procedure for the synthesis of compounds 7a, b from essential oil of thyme (Thymus vulgaris)).
- The manuscript text does not include a description of the preparation of compounds 9 and 10.
Author’s response: The description can be found in the 2. Results and Discussion section (2.1.2. Experimental results of oxidative rearomatization) and 3. Materials and Methods section (3.1.3. General procedure for the oxidation of thymol and carvacrol containing benzothiazolines 7a-c, 8a-c toward aromatized products 9 and 10).
- I also suggest removing the numbering in Figure 3.
Author’s response: It is removed in the revised version of the manuscript.
- What is the rationale behind the selected concentration range for determining the sun protection factor (SPF) in vitro? What solvent was used to prepare the concentrations of the studied compounds?
Author’s response: The in vitro sun protection factor (SPF) of some of the new synthetic analogues of 2-phenyl-1H-benzimidazole-5-sulfonic acid (PBSA), was investigated. In the conducted in vitro studies, PBSA was chosen as the reference standard substance. The maximum concentration of PBSA approved by the Food and Drug Administration (FDA) is 148 mM. In commercial sunscreens, UVB filter PBSA concentrations often range between 74 mM and 148 mM. The data for in vitro or in vivo tests of SPF factor of studied compounds were not found in the literature. Therefore, in the current studies, we chose concentrations much lower than those allowed for PBSA. We selected the following working concentrations for this investigation - from 1 mM to 0.005 mM. The concentrations are prepared in 96% ethanol as an appropriate solvent for the examination.
- I propose adding references to the synthesis methods for compounds 2 and 11 to the experimental section since their synthesis is mentioned in the text but is not described in the results or the experimental section.
Author’s response: The compound 11, PBSA was purchased commercially. The additional information was added in Experimental section (3. Results and Methods, 3.1.1. General information). The reference from our previous study describing the synthesis and analysis of compound 2 (BT-PhOH hybrid) was also introduced in Experimental section (3. Results and Methods, 3.1.1. General information).
- What solvent was used to prepare the concentrations of the studied compounds in the investigation of their antimicrobial properties? Was the stability of the studied compounds in the solvent used for antimicrobial activity studies evaluated?
Author’s response: The solvent used for the antimicrobial properties investigations is dimethyl sulfoxide. Solutions of the various substances were freshly prepared immediately before testing for antimicrobial activity. Previous studies established the stability of the obtained solutions through structural analyses of the substances, confirming the preservation of their chemical structure.
- The authors of the manuscript should unify the description of the 1H^1 \mathrm{H}1H-NMR spectra. When describing the spectra of the obtained compounds, specifically doublet, triplet, and multiplet signals, the authors sometimes use the range of signal values and sometimes use the mean value. For example, for compound 7b, the signal is described as 7.21 (d, J=7.6J = 7.6J=7.6, 1H, -CH-Benzothiazole), while for compound 7c, it is 4.90–4.99 (d, 2H, -COOCH2_22Cl).
Author’s response: The NMR descriptions are corrected in the revised version of the manuscript and supplementary material.
- In the experimental section, the 1H^1 \mathrm{H}1H-NMR spectra should list the proton signals separated by commas (compounds 7c and 8c).
Author’s response: The proton signals in 1H-NMR spectras are correctly separated by comas in the revised version of the manuscript.
- There are errors in the compound names: (4-(benzo[d]thiazol-2-yl)-2-isopropyl-5-methylphenol) and (3-(benzo[d]thiazol-2-yl)-5-isopropyl-2-methylphenol), specifically in the term "isopropyl." Corrections are necessary.
Author’s response: The compound names are carefully corrected in the revised version of the manuscript.
The all remarks are taken into account. In this regard, the additional corrections of manuscript are made and marked (track changes).
Reviewer 2 Report
Comments and Suggestions for Authors
The manuscript titled „ New benzothiazole-monoterpenoid hybrids as multifunctional 2 molecules with potential applications in cosmetics “ investigated and assessed the synthesis and spectral characterization of 2-substituted benzothiazole derivatives, incorporating natural monoterpenoids, such as thymol and carvacrol, found in thyme essential oil.
The manuscript is well-written and organized, the language is precise and understandable. The introduction section is comprehensive and covers a wide range of references of recent date. Experimental procedures are explained in detail.
This study is very interesting and can contribute to the current knowledge. The core of this research has some novelty and significance. However, the manuscript needs minor revisions. Some issues need to be addressed for better understanding before its publication:
Introduction section. The authors need to explicitly state the novelty of this approach and, hence, the sentence „In the literature, data on O-alkylation, O-acylation [31,32] and C-acylation [10] for the synthesis of thymol and carvacrol derivatives was successfully found, however, this is not applicable to C-alkylation processes.” should be reformulated to be more clear; Are all of the synthesized compounds new?
Figure 1: caption must be more comprehensive;
Results and discussion section.
Table 1: Abbreviations BT and DDQ should be explained since it is the first place of their appearance;
Lines 194-195: It would be useful for better understanding to designate compounds from Table 1 to which this statement is referred to;
Line 215: “compounds 9 and 10” instead of “compounds 9,10”;
Figure 4: In this figure, a comma should be replaced with a point;
Conclusion. It is too long with bulky sentences. Need to be rewritten to be more concise.
Line 767: “synthesis” instead of “synthsis”
Author Response
Response to Reviewer 1 Comments
Dear Reviewer, thank you for your effort to provide timely review and decision on our manuscript.
Reviewer 2 Comments
The manuscript titled „New benzothiazole-monoterpenoid hybrids as multifunctional 2 molecules with potential applications in cosmetics“ investigated and assessed the synthesis and spectral characterization of 2-substituted benzothiazole derivatives, incorporating natural monoterpenoids, such as thymol and carvacrol, found in thyme essential oil.
The manuscript is well-written and organized, the language is precise and understandable. The introduction section is comprehensive and covers a wide range of references of recent date. Experimental procedures are explained in detail.
This study is very interesting and can contribute to the current knowledge. The core of this research has some novelty and significance. However, the manuscript needs minor revisions.
Author’s comments: Thank you for the positive point of view and scientific discussion on our work.
Some issues need to be addressed for better understanding before its publication:
Introduction section.
„The authors need to explicitly state the novelty of this approach and, hence, the sentence „In the literature, data on O-alkylation, O-acylation [31,32] and C-acylation [10] for the synthesis of thymol and carvacrol derivatives was successfully found, however, this is not applicable to C-alkylation processes.” should be reformulated to be more clear; Are all of the synthesized compounds new?“
Author’s comments: Thank you for the constructive remark. The sentence is clearly reformulated. We represent a convenient and easily accessible two-step approach for the synthesis of novel compounds. The all benzothiazole-monoterpenoid hybrids (7a-c, 8a-c, 9 and 10) are new.
„Figure 1: caption must be more comprehensive;“
Author’s comments: The caption and information of the Figure 1 are corrected in the revised version of the manuscript.
Results and discussion section.
“Table 1: Abbreviations BT and DDQ should be explained since it is the first place of their appearance;”
Author’s comments: The abbreviations are explained in their first appearance in the revised text of the manuscript.
“Lines 194-195: It would be useful for better understanding to designate compounds from Table 1 to which this statement is referred to;”
Author’s comments: It is done in corrected version of the manuscript.
“Line 215: “compounds 9 and 10” instead of “compounds 9,10”;”
Author’s comments: It is corrected in the revised version of the manuscript.
“Figure 4: In this figure, a comma should be replaced with a point;”
Author’s comments: It is modified in the revised version of the manuscript.
“Conclusion. It is too long with bulky sentences. Need to be rewritten to be more concise.”
Author’s comments: Some of the bulky sentences are rewritten of the revised version of the manuscript.
“Line 767: “synthesis” instead of “synthsis”
Author’s comments: Thank you for the careful observation. It is corrected in the revised manuscript text.
The all remarks are taken into account. In this regard, the additional corrections of manuscript are made and marked (track changes).
Reviewer 3 Report
Comments and Suggestions for Authors
Recommendation: Publish after major revisions noted.
Comments:
This manuscript from Desislava Kirkova et. al developed an effective synthetic method for new benzothiazole-monoterpenoid hybrids and their potential applications in cosmetics were well studied. This article highlights discusses benzothiazole monoterpenoid hybrids as new hit scaffold for multifunctional molecules in cosmetics. However, there are some questions while I read the current manuscript. Thus I recommended a minor revision is required before this work has been published on Molecules.
(1) In Scheme 1, the yield of each step is suggested to added for professionalism and readability.
(2) Do the authors optimize the temperature for the 1H NMR of the synthesis of benzothiazole-monoterpenoids hybrids?
(3) In this study, the authors have provided the in-silico predictions of benzothiazole-based hybrids toxicity. However, the cosmetic toxicology experiments such as Skin irritation test and Acute toxicity test are suggested in the text.

Author Response
Response to Reviewer 3 Comments
Dear Reviewer, thank you for your effort to provide timely review and scientific conclusion on our manuscript.
Reviewer 3 Comments
Recommendation: Publish after major revisions noted.
Author’s comments: The major revisions are made in the revised text of the manuscript and supplementary information.
Comments: This manuscript from Desislava Kirkova et. al developed an effective synthetic method for new benzothiazole-monoterpenoid hybrids and their potential applications in cosmetics were well studied. This article highlights discusses benzothiazole monoterpenoid hybrids as new hit scaffold for multifunctional molecules in cosmetics. However, there are some questions while I read the current manuscript.
Author’s comments: Thank you for the positive point of view and scientific discussion on our work. The all remarks are taken into account. In this regard, the additional corrections of manuscript are made and marked (track changes).
Thus I recommended a minor revision is required before this work has been published on Molecules.
(1) In Scheme 1, the yield of each step is suggested to added for professionalism and readability.
Author’s comments: Thank you for the recommendation. The yields of each step are added in the revised version of the Scheme 1 in manuscript.
(2) Do the authors optimize the temperature for the 1H NMR of the synthesis of benzothiazole-monoterpenoids hybrids?
Author’s comments: To average out the rotamers reaching adequate assignment of peaks and structure determination, the spectra were measured at 80°C. This optimisation led to averaged signals successfully confirming the expected signals for the structures of the newly obtained products. A short discussion is added to the revised text of the manuscript.
(3) In this study, the authors have provided the in-silico predictions of benzothiazole-based hybrids toxicity. However, the cosmetic toxicology experiments such as Skin irritation test and Acute toxicity test are suggested in the text.
Author’s comments: The toxicity of a newly synthesized compound is assessed not only by its potential impact on human health, but also on the environment. In silico prediction for the toxicity of benzothiazole-based hybrids provides computer data that identifies possible risks and draws attention to potential risks hidden by the newly synthesized compounds. Data for additional cosmetic toxicological in silico analyses, such as skin irritation, determine whether the compound may cause irritation on contact with the skin. The results of this test are important for assessing the safety of direct application of the product to the skin. These projections serve as an initial assessment stage that allows for more efficient planning of subsequent experimental studies. Nevertheless, to make a final conclusion about the properties of these compounds, additional in vitro and in vivo tests are necessary. Combining in silico analyses with experimental in vitro and in vivo tests provides a more complete and reliable assessment of toxicity.
Reviewer 4 Report
Comments and Suggestions for Authors
Below, there is the review of the manuscript (molecules-3365832-peer-review-v2) entitled “New benzothiazole-monoterpenoid hybrids as multifunctional molecules with potential applications in cosmetics”
The work presented by the Authors (i.e., Desislava Kirkova , Yordan Stremski, Maria Bachvarova , Mina Todorova , Bogdan Goranov, Stela Statkova-Abeghe , Margarita Docheva) is a continuation of their research on benzothiazole-monoterpenoid hybrid derivatives. In the current study, the Authors synthesized some benzothiazole-monoterpenoid hybrid derivatives containing thymol and carvacrol moieties and tested them in some biological assays including antimicrobial ones.
My remarks are the following:
1. There is no confirmation of the purity of tested compounds (such as combustion analysis) in biological assays. It should be emphasized that HRMS spectroscopy (as some people believe) is not used to determine the purity of chemical compounds.
2. The results of the antibacterial test presented in Table 7 should be processed accordingly. MIC50 and MIC90 should be determined and absolutely presented in the form of molar concentrations, which is obvious if only because of the significant differences between the molar masses of the tested compounds and the reference compounds.
3. Why did the authors not include derivatives 7c and 8c in their biological studies?
4. I suggest Table 3 be moved to supplementary, or at least remove SMILES notation completely unnecessary in this work.
5. Theoretical predictions using known computational techniques applied in silico studies should absolutely be verified by appropriate experimental studies because without this they have little value.
6. In the supplementary visualizations of IR spectra, please provide the numerical values of the band maxima.
In conclusion, the manuscript should be thoroughly revised and supplemented, and then it can be considered for acceptance for publication.
Author Response
Response to Reviewer 4 Comments
Dear Reviewer, thank you for your effort to provide timely review and scientific recommendations on our manuscript.
Reviewer 4 Comments
Below, there is the review of the manuscript (molecules-3365832-peer-review-v2) entitled “New benzothiazole-monoterpenoid hybrids as multifunctional molecules with potential applications in cosmetics”
The work presented by the Authors (i.e., Desislava Kirkova, Yordan Stremski, Maria Bachvarova , Mina Todorova , Bogdan Goranov, Stela Statkova-Abeghe , Margarita Docheva) is a continuation of their research on benzothiazole-monoterpenoid hybrid derivatives. In the current study, the Authors synthesized some benzothiazole-monoterpenoid hybrid derivatives containing thymol and carvacrol moieties and tested them in some biological assays including antimicrobial ones.
Author’s comments: Thank you for the positive point of view and scientific discussion on our work.
My remarks are the following:
- There is no confirmation of the purity of tested compounds (such as combustion analysis) in biological assays. It should be emphasized that HRMS spectroscopy (as some people believe) is not used to determine the purity of chemical compounds.
Author’s comments: Thank you for the advice. Analytically pure samples of only para-substituted products were isolated by column chromatographic separation on silica using a mixture of petroleum/diethyl ether as eluents. The purity was monitored by thin-layer chromatography (TLC) and confirmed by 1H-, 13C-NMR experiments. The biological activity of the tested compounds was examined without the need of their further purification.
- The results of the antibacterial test presented in Table 7 should be processed accordingly. MIC50 and MIC90 should be determined and absolutely presented in the form of molar concentrations, which is obvious if only because of the significant differences between the molar masses of the tested compounds and the reference compounds.
Author’s comments: Dear reviewer thank you for the constructive advice. In determining the antimicrobial activity potential of the newly synthesized compounds, as well as the reference substances, we selected pathogens in accordance with the standards that describe the main microbiological indicators of cosmetic products. Since the reference substances and some analogues exhibited almost no activity against the test microorganisms, except for C. albicans NBIMCC 74, the authors concluded that MIC50 and MIC90 could not be calculated. This can only be defined for analogues 7a, 7b, and 8a, which exhibit antimicrobial activity against all six tested pathogens. The results in Table 7 show that the MIC50 and MIC90 for these substances are identical at 15 µg/mL.
The initial idea in determining the antimicrobial activity of the substances was to establish whether they would act on potential pathogens in the cosmetic product to ensure safety. In further studies, we will test the antimicrobial activity against more strains of the same species to determine the strain-specific action of the analogues. Additionally, more precise determination of MIC50 and MIC90 for the strains of individual species will be sought, as well as ways to functionalize the chemical structure of the compounds to enhance their biological activity and properties. The concentration of the substances tested for antimicrobial activity is expressed in µg/mL to allow comparability between the different substances. The molecular weight of thyme essential oil, which is one of the reference substances, cannot be determined due to its multicomponent composition. Furthermore, it was found in most of the reviewed literature that the majority of authors indicate the concentration of synthetic substances with antimicrobial activity in this manner.
- Why did the authors not include derivatives 7c and 8c in their biological studies?
Author’s comments: Thank you for the remark. Due to the higher chlorine atom content in these molecules, we considered that they would not possess high biological activity. A similar conclusion was reached by a research group that recently published various Troc-containing Camalexin analogues obtained using our two-step approach [Liao, A.; Li, L.; Wang, T.; Lu, A.; Wang, Z.; Wang, Q. Discovery of Phytoalexin Camalexin and Its Derivatives as Novel Antiviral and Antiphytopathogenic-Fungus Agents. Journal of Agricultural and Food Chemistry 2022, 70(8), 2554–2563. https://doi.org/10.1021/acs.jafc.1c07805].
Nevertheless, we believe that these examples correctly complement our proposed approach for the facile synthesis of aromatic thymol and carvacrol containing hybrids.
- I suggest Table 3 be moved to supplementary, or at least remove SMILES notation completely unnecessary in this work.
Author’s comments: Thank you for the constructive remark. The Table 3 is correctly modified in the revised version of the manuscript.
- Theoretical predictions using known computational techniques applied in silico studies should absolutely be verified by appropriate experimental studies because without this they have little value.
Author’s comments: Thank you for the correct observations. The in silico studies only give us a good lead for further appropriate experimental studies for additional verification. This will be our point for our future investigations and examinations.
- In the supplementary visualizations of IR spectra, please provide the numerical values of the band maxima.
Author’s comments: The numerical values of the band maxima of IR spectra are provided in the revised supplementary material.
In conclusion, the manuscript should be thoroughly revised and supplemented, and then it can be considered for acceptance for publication.
Author’s comments: The major revisions and supplementations are made in the revised text of the manuscript and supplementary information. The all remarks are taken into account. In this regard, the additional corrections of manuscript are made and marked (track changes).
Reviewer 5 Report
Comments and Suggestions for Authors
The manuscript submitted by Stremski and co-workers describes the synthesis of benzothiazole-monoterpenoid hybrids and the investigation by UV spectroscopy to assess potential use as sunscreen filters. In addition, some biological activities have been also analysed. Overall, the manuscript is interesting. There are some issues that should be addressed before publication.
In particular, table 1 could be split in two tables.
More details about how the authors determined the sun protection factor (SPF) using the Mansur equation should be given. The explanation reported in the section 3.1.5. is not clear enough. This should be discussed in the main text as well as the significance of the reported results.
I do not catch the relationship between in silico predicted properties of the amidoalkylated compounds 7 336 and 8 (Tables 4 and 5) and the lack of conjugated double bonds led to the hypothesis that 337 the compounds might not be effective UV filters (lines 336 and 337).
Even though wild thyme, thymol, and carvacrol are used as antimicrobial agents in various foods and cosmetics [ ref 5,53], the authors have not found antimicrobial activity in their investigation. This should be discussed.
In the determination of the radical scavenging activity, the hybrid compounds of thymol and carvacrol 7a,b, and 8a,b, containing a benzothiazole fragment in their molecule, showed extremely low radical scavenging activity. The authors stated that this was expected, but these results have not been commented. This is counterintuitive since the natural products show radical scavenging activity.
Regarding the supplementary materials, the graphical quality of the spectra should be improved. The integrals and pick peaking overlap with the NMR signals.
Author Response
Response to Reviewer 5 Comments
Dear Reviewer, thank you for your effort to provide timely review and scientific conclusion on our manuscript.
Reviewer 5 Comments
“The manuscript submitted by Stremski and co-workers describes the synthesis of benzothiazole-monoterpenoid hybrids and the investigation by UV spectroscopy to assess potential use as sunscreen filters. In addition, some biological activities have been also analysed. Overall, the manuscript is interesting.”
Author’s comments: Thank you for the positive point of view and scientific discussion on our work.
There are some issues that should be addressed before publication.
“In particular, table 1 could be split in two tables.”
Author’s comments: Table 1 is divided into two separate tables, reflected in the manuscript.
“More details about how the authors determined the sun protection factor (SPF) using the Mansur equation should be given. The explanation reported in the section 3.1.5. is not clear enough. This should be discussed in the main text as well as the significance of the reported results.”
Author’s comments: Thank you for the scientific remarks. Corrections are reflected in sections 2.2.2. and 3.1.7. Table 7 shows the results for 0.0312 mM solution of compound 10 in 96% ethanol. For all other concentrations and synthetic compounds, the presentation of the results is not represented in this tabular form. Because a lot of experimental work has been done, which is voluminous and would be difficult to present in a manuscript. The results obtained for the remaining concentrations of 2-arylbenzazole compounds - 2, 9, 10 and 11 were systematized and summarized and presented in Table 8.
“I do not catch the relationship between in silico predicted properties of the amidoalkylated compounds 7 336 and 8 (Tables 4 and 5) and the lack of conjugated double bonds led to the hypothesis that 337 the compounds might not be effective UV filters (lines 336 and 337).”
Author’s comments: After correction, Tables 4 and 5 are replaced with Tables 5 and 6. Minimal corrections have been made. This article aims to synthesize new benzothiazole-monoterpenoid hybrids as multifunctional molecules with potential applications in cosmetics. In connection with the aim, the spectral properties and the SPF factor was calculated using a validated method of the amidoalkylated hybrids were investigated. The absence of a second aromatic ring in the molecule and the presence of an amide group significantly affects the spectral properties. It is known from the literature that the lack of spliced double bonds in the molecules significantly reduces the effectiveness of UV filters. The lack of such a structure limits the ability of molecules to absorb and neutralize UV rays, which is an exceptional characteristic. A plateau was found in the compound 7а spectrum with no expressed maximum absorption in the UVB region (Figure 4. UV spectra of compound: 7a (purple line). That is why we carried out oxidative aromatization and successfully obtained aromatic compounds 9 and 10. Tables 5 and 6 present the results of the predicted values after an in silico analysis: The results obtained for compounds 7 and 8 show that they are mutagenic and at very low concentrations are toxic. This makes them ineffective for cosmetics. The lack of absorption in the UVB region (wavelengths between 290 and 320 nm) means that substances cannot effectively protect the skin from these particular wavelengths of ultraviolet radiation.
“Even though wild thyme, thymol, and carvacrol are used as antimicrobial agents in various foods and cosmetics [ref 5,53], the authors have not found antimicrobial activity in their investigation. This should be discussed.”
Author’s comments: Thank you for the constructive remark. Тhe probable cause is mentioned in the revised text of the manuscript. In this regard, antimicrobial activity depends on the extraction conditions, which significantly determine the component composition and, consequently, the biological activity. The antimicrobial effect also depends on the initial concentration of the test microorganisms and their sensitivity. In literature source [53], the concentration of test microorganisms is approximately 10³ cfu/cm³, which does not correspond to the McFarland 0.5 standard (1.5 × 10⁸ cfu/cm³) accepted as the standard concentration for testing antimicrobial activity. The reference substances are commercial products, not obtained by us, making it difficult to compare the results with data from other authors.
“In the determination of the radical scavenging activity, the hybrid compounds of thymol and carvacrol 7a,b, and 8a,b, containing a benzothiazole fragment in their molecule, showed extremely low radical scavenging activity. The authors stated that this was expected, but these results have not been commented. This is counterintuitive since the natural products show radical scavenging activity.”
Author’s comments: The radical-scavenging activity of the new thymol and carvacrol hybrid compounds 7a, b and 8a, b was investigated using two methods – DPPH and ABTS. The results for the thymol hybrids by the ABTS method are lower than the starting carvacrol but close to thymol, while the results for the carvacrol hybrids are almost twice as low as the starting thymol and carvacrol. The obtained data show a difference between the activities of the new thymol and carvacrol hybrids, and the results are discussed.
In contrast, the radical-scavenging activity of the new thymol and carvacrol hybrids by the DPPH method shows extremely low activity of the synthetic molecules compared to the starting thymol and carvacrol. The ability to scavenge free radicals varies depending on the structure of the radical, which is why we chose to study the antioxidant activity of the new hybrids using two different methods. In our earlier studies, a correlation between the structure of 2-phenyl benzothiazole compounds by the ABTS method was observed, while no such correlation was found by the DPPH method. Therefore, the low values obtained by the DPPH method were expected.
Determining the radical-scavenging activity of the studied compounds using ABTS free radicals is more suitable than the DPPH radical.
“Regarding the supplementary materials, the graphical quality of the spectra should be improved. The integrals and pick peaking overlap with the NMR signals.“
Author’s comments: The graphical quality of the NMR spectra in supplementary materials is clearly improved for all of the synthesized compounds.
The all remarks are taken into account. In this regard, the additional corrections of manuscript are made and marked (track changes).
Round 2
Reviewer 3 Report
Comments and Suggestions for Authors
All mentioned concerns have been addressed by the authors, now the manuscript can be accepted.
Author Response
Response to Reviewer 3 Comments
Dear Reviewer, thank you for your effort to provide a timely response and scientific decision on our manuscript.
Reviewer 3 Comments
Comments and Suggestions for Authors
All mentioned concerns have been addressed by the authors, now the manuscript can be accepted.
Author’s comments: Dear Reviewer, thank you for your comments and recommendations. Thanks to them, the reviewed article has greater coherence, and scientific accuracy suitable for Molecules.

Reviewer 4 Report
Comments and Suggestions for Authors
Below, there is the re-review of the manuscript (molecules-3365832-peer-review-v3) entitled “New benzothiazole-monoterpenoid hybrids as multifunctional molecules with potential applications in cosmetics”
Unfortunately, the authors have not properly corrected important deficiencies. Analyses should be performed to confirm purity in a quantitative form (such as combustion analysis or quantitative chromatographic analysis) not qualitative.
The results of the antibacterial test presented in Table 7 should be determined and absolutely presented in the form of molar concentrations. This is obvious because of the significant differences between the molar masses of the tested compounds and the reference compounds.
I believe that in this form the manuscript is not suitable for publication in Molecules.
Author Response
Response to Reviewer 4 Comments
Dear Reviewer, thank you for your effort to provide a timely response and scientific recommendations on our manuscript.
Reviewer 4 Comments
Comments and Suggestions for Authors
Below, there is the re-review of the manuscript (molecules-3365832-peer-review-v3) entitled “New benzothiazole-monoterpenoid hybrids as multifunctional molecules with potential applications in cosmetics”
Unfortunately, the authors have not properly corrected important deficiencies. Analyses should be performed to confirm purity in a quantitative form (such as combustion analysis or quantitative chromatographic analysis) not qualitative.
Author’s comments: Dear Reviewer, the samples of the obtained compounds were isolated by column chromatographic separation and the yields are calculated in a quantitative form according the procedure in our recently published work in Molecules [https://doi.org/10.3390/molecules30010107]. Thus, we applied the method using small quantities of the starting reagents. In the future, we plan to scale up the reaction to obtain gram-scale quantities of the carvacrol-containing compounds. At that time, additional analyses such as combustion analysis or quantitative chromatographic analysis will be prioritized and applied.
We believe that, for the context of this work the investigations about the synthesis of the novel hybrid molecules and their bioactivity potential are clearly visualized. I this regard, the reviewed form of the manuscript and supplementary materials are correctly presented.
The results of the antibacterial test presented in Table 7 should be determined and absolutely presented in the form of molar concentrations. This is obvious because of the significant differences between the molar masses of the tested compounds and the reference compounds.
Author’s comments: The results in Table 7 connected to the sample content are recalculated according to the different molar masses of the tested compounds and presented in μmol.
I believe that in this form the manuscript is not suitable for publication in Molecules.
Author’s comments: Dear reviewer, thank you for the constructive opinion. We hope that after the corrections and innovations that we made, you will approve the text of the revised manuscript and its publication in this form.
